# Tuberculin skin test positivity among HIV-infected alcohol drinkers on antiretrovirals in south-western Uganda

Winnie R. Muyindike[1,2]*, Robin Fatch[3], Debbie M. Cheng[4], Nneka I. Emenyonu[3], Christine Ngabirano[5], Julian Adong[6], Benjamin P. Linas[7], Karen R. Jacobson[7], Judith A. Hahn[3,8]

1 Department of Internal Medicine, Faculty of Medicine, Mbarara University of Science and Technology, Mbarara, Uganda, 2 Department of Internal Medicine, Mbarara Regional Referral Hospital, Mbarara, Uganda, 3 Department of Medicine, University of California San Francisco, San Francisco, California, United States of America, 4 Department of Epidemiology, Boston University School of Public Health, Boston, Massachusetts, United States of America, 5 Grants Office, Directorate of Research and Post graduate Studies, Mbarara University of Science and Technology, Mbarara, Uganda, 6 Department of Paediatrics, Faculty of Medicine, Mbarara University of Science and Technology, Mbarara, Uganda, 7 Section of Infectious Diseases, Boston Medical Center, Boston, Massachusetts, United States of America, 8 Department of Epidemiology and Biostatistics, University of California San Francisco, San Francisco, California, United States of America

* wmuyindike@gmail.com

**Data Availability Statement:** All relevant data are within the manuscript and its Supporting Information files. This data set is also available within the URBAN ARCH data repository

## Abstract

### Background

Tuberculosis (TB) is the leading cause of death among people living with HIV (PLWH), and current evidence suggests that heavy alcohol users have an increased risk of developing TB disease compared to non-drinkers. Not known is whether the increased risk for TB disease among alcohol users may reflect higher rates of latent TB infection (LTBI) among this population. We assessed the latent TB infection prevalence based on tuberculin skin testing (TST) and examined association with current alcohol use among HIV-infected persons on antiretroviral therapy (ART) in south-western Uganda.

### Methods

We included PLWH at the Mbarara Regional Hospital HIV clinic, who were either current alcohol consumers (prior 3 months) or past year abstainers (2:1 enrolment ratio). Participants were recruited for a study of isoniazid preventive therapy for LTBI. TST was performed using 5 tuberculin units of purified protein derivative. The primary outcome was a positive TST reading (≥5mm induration), reflecting LTBI. We used logistic regression analyses to assess the cross-sectional association between self-reported current alcohol use and a positive TST.

### Results

Of the 295 of 312 (95%) who returned for TST reading, 63% were females and 63% were current alcohol drinkers. The TST positive prevalence was 27.5% (95% confidence interval [CI]: 22.6% - 32.9%). The odds of a positive TST for current alcohol users compared to

(http://sites.bu.edu/urbanarch/resources/data-sample-repository/).

**Funding:** This study was supported by Grants from National Institute on Alcohol Abuse and Alcoholism (NIAAA): U01AA020776 and K24 AA022586 (PI: JH); U24AA020779 (PI: DC) and NIH/NIAID P30AI042853 (support to DC). NIAAA provided support in the form of effort-based salaries for the authors WM, RF, DC, NE, CN, JA, BL, KJ, JH. The National Institute of Allergy and Infectious Diseases (NIAID) provided support to DC. The funders did not have any role in the study design, data collection and analysis, decision to publish, nor preparation of the manuscript. Author DC serves on Data Safety and Monitoring Boards for Janssen. The specific roles of these authors are articulated in the 'author contributions' section.

**Competing interests:** Author DC serves on Data Safety and Monitoring Board for Janssen. The other authors have no conflicts of interest to declare. There are no patents, products in development or marketed products to declare. This does not alter our adherence to PLOS ONE policies on sharing data and materials.

abstainers was 0.76 (95% CI: 0.41, 1.41), controlling for gender, age, body mass index, history of smoking, and prior unhealthy alcohol use.

## Conclusions

The prevalence of LTBI among PLWH on ART in south-western Uganda was moderate and LTBI poses a risk for future infectious TB. Although alcohol use is common, we did not detect an association between current drinking or prior unhealthy alcohol use and LTBI. Further studies to evaluate the association between LTBI and different levels of current drinking (heavy versus not) are needed.

## Introduction

Globally, tuberculosis (TB) is the leading cause of mortality among people living with HIV (PLWH) [1]. In 2018, there were an estimated 0.9 million new cases of TB disease globally amongst PLWH [1] and an estimated quarter of the world's population is infected with TB (latent TB infection, LTBI) [1–3]. Progression to active TB, either through primary progression or reactivation of LTBI is 20–30 times higher in PLWH than those without [2]. In sub-Saharan Africa (SSA), LTBI in the general population is estimated to range from 34–75% [4–6], with no difference found by HIV status in recent studies [5, 6]. However, TB preventive therapy is recommended for PLWH in high burden TB settings because of the high risk of progression to active disease [1, 2, 7]. In Uganda, the current TB incidence rate is estimated at 200 per 100,000 population. This is lower than that of South Africa, where the TB incidence rate is estimated at 520 per 100,000 but is higher than incidence reported in low TB burdened areas of less than 100 per 100,000 population [1].

Alcohol use is a leading risk factor for global disease burden and health loss [8], including TB [9], and heavy alcohol consumption is associated with a 2.5-fold increased risk of active TB disease compared to individuals who did not drink alcohol [9, 10]. This elevated TB disease risk may be attributed to a higher LTBI prevalence among people who are heavy alcohol consumers, possibly due to social factors such as frequenting bars that may increase exposure opportunity [11–13]. Alternately, excess TB disease burden may reflect a faster rate of progression from primary infection or LTBI to active disease, due to impaired immune function in heavy alcohol users [11, 14]. It is important to know if alcohol drinkers in TB- and TB/HIV-burdened areas have a higher prevalence of LTBI than non-drinkers, given that the recommendations for isoniazid preventive therapy (IPT) in low resource settings caution against IPT use in regular and heavy alcohol users [7, 15], although IPT reduces all-cause mortality among HIV-infected persons by 32–62% [16, 17] and reduces the incidence of TB reactivation beyond the benefit of using ART alone [18].

Uganda has a prevalence of HIV at 6.2% [19], and heavy alcohol consumption is estimated at 25% among PLWH in those settings [20–23]. While some literature cites no association between alcohol use and TB infection [6, 24], there is little detail specifically targeting PLWH on ART. Hence, we sought to: i) Estimate the prevalence of LTBI confirmed using the tuberculin skin test (TST) among PLWH on ART and ii) Evaluate the association between current alcohol use and TST positivity. We hypothesized that participants with self-reported current alcohol consumption have higher odds of having a positive TST.

## Methods

### Study setting and population

We collected data during the screening visit for the Alcohol Drinkers' Exposure to Preventive Therapy for TB (ADEPTT) Study, which is part of the Uganda Russia Boston Alcohol Network for Alcohol Research Collaboration on HIV/AIDS (URBAN ARCH). The primary aim of the ADEPTT study is to estimate the rate of hepatotoxicity to isoniazid (6 months INH) treatment among adults ($>$ = 18 years) who are HIV/LTBI co-infected and are current (prior 3 months) alcohol drinkers and to compare that rate to hepatotoxicity among non-drinkers (abstaining at least one year). We screened PLWH who participated in prior URBAN ARCH studies of alcohol use in the Immune Suppression Syndrome (ISS) clinic of Mbarara Regional Referral Hospital (MRRH) in south-western Uganda from 2010–2017 [25, 26].

Eligibility criteria was based on the main ADEPTT study and included: 1) age $\geq$ 18 years, 2) fluency in English or Runyakole (the local language), 3) HIV-infected on a non-nevirapine containing ART regimen for at least six months, 4) living within 2 hours travel time from the clinic with no plans to move, 5) no evidence of active TB infection based on WHO symptoms criteria, 6) no prior use of TB medicines for treatment and/or prevention of TB, 7) being either a self-reported current (prior 3 month) drinker or abstainer (at least one year since drinking), 8) having alanine aminotransferase [ALT] and aspartate aminotransferase [AST] $\leq$ 2x of the upper limit of normal (ULN), 9) women who were pregnant.

### Study procedures and variables collected

The data for the analyses came from three sources: the ADEPTT screening process, prior URBAN ARCH research data, and the ISS clinic database. Data were linked via study and clinic identification numbers.

**Tuberculin Skin Testing (TST).** The TST was conducted by placing 0.1 ml containing 5 tuberculin units of purified protein derivative (PPD) (Tubersol; Sanofi Pasteur Limited, Toronto, Ontario, Canada) on participants' forearms using the Mantoux method. The indurations were read by trained clinical research assistants (CRAs) and cross-checked by another CRA using a TST tape measure between 48 and 72 hours after placement. TST may have reduced sensitivity in immune-compromised individuals such as those with HIV infection [27], nonetheless, in this study, an induration of 5mm or greater was considered a positive TST result (the cut off recommended for those with HIV) [28]. WHO recommends using either TST or interferon-gamma release assays (IGRA) [1] for latent TB detection.

**Outcome variable.** The outcome measure was a positive TST result during ADEPTT screening, as described above.

**Independent variable.** The primary predictor of interest was current alcohol consumption, defined as self-reported alcohol use within 3 months prior to the TST. This variable was obtained by self-report during the ADEPTT screening procedures.

**Covariates.** Gender (male, female) was collected during ADEPTT screening. Data from prior URBAN ARCH study visits included lifetime history of smoking (never versus ever) and "prior unhealthy alcohol use". Prior unhealthy alcohol use was determined using the Alcohol Use Disorders Identification Test—Consumption (AUDIT-C), which was modified to ask participants to answer the AUDIT-C questions based on the period in their life during which they drank the most. Those meeting AUDIT-C cut-offs for this prior drinking period ($>$ = 3 for women and $>$ = 4 for men, [29]) were considered prior unhealthy drinkers. These measurements from prior URBAN ARCH study visits were collected 3–5 years before the screening visit for ADEPTT.

Date of birth, marital status, ART initiation date, body mass index (BMI) and most recent HIV viral load were linked from ISS clinic data. Age and duration of ART at TST placement were calculated from these data. Age was categorized as <35 or ≥35 years; BMI was categorized as low (<18.5), normal (18.5–24.9), overweight (25–29.9), and obese (≥30); viral load was categorized as ≤1000 and >1000 copies/ml.

## Statistical methods

We conducted descriptive analyses of all variables, overall and by self-reported current alcohol use. We calculated frequencies and proportions for categorical variables; we calculated means and standard deviations (std dev) and medians with interquartile ranges (IQR) for continuous variables. We ran unadjusted logistic regression models and multivariable logistic regression models to assess the association between current alcohol use and a positive TST. The multivariable model controlled for age, gender, BMI, smoking history, and prior unhealthy alcohol use. These covariates were chosen for inclusion in the multivariable analysis a priori based on literature of the risk factors that may increase TB disease cases [1, 30]. We also conducted exploratory analyses to assess whether the association between alcohol use and TST positivity differed by gender, and present multivariable results stratified by gender. We also compared the distributions of gender, age, BMI and alcohol use of the persons that were included and those screened but excluded either because they did not meet the main study inclusion criteria or did not return for TST reading on time.

## Ethical considerations

Written informed consent was sought from participants before screening. The study activities were approved by the Ethics review boards of Mbarara University of Science and Technology (#11/10-16), The Uganda National Council of Science and Technology (# HS 2183), University of California San Francisco (# 16–19093), and Boston University (# H-35809).

## Results

### Characteristics of the study participants

We approached 418 prior URBAN ARCH participants for ADEPTT Study screening; 106 persons were excluded for reasons that were not mutually exclusive, including: being on nevirapine (n = 31), having had previous active TB (n = 14), having previously taken TB medicines for LTBI (n = 25), having consumed alcohol in the past year but not the past 3 months (n = 8), having elevated baseline AST and/or ALT results (n = 18), living out of catchment area (n = 15), not on ART equal or more than 6 months (n = 1), pregnant (n = 2), not cleared of active TB (n = 2) and declining participation (n = 7). 312 participants received a TST. Seventeen participants did not return for PPD reading within 72 hours, leaving 295 for this analysis (Fig 1). A comparison between the 295 persons included and the 123 persons excluded from the analyses did not show any difference by alcohol consumption, age, gender, or BMI.

Of the 295 who met inclusion criteria, 63% were female, median age was 38 years (IQR: 32 to 45), 56% were married, 92% had been on ART for at least one year, 97% were virally suppressed (≤1000 RNA copies), and median BMI was 23.3 (IQR: 20.8 to 27.5) (Table 1). Seventeen percent reported ever smoking in their lifetime. Approximately two-thirds (63%) were self-reported current alcohol drinkers, and a similar proportion (57%) were prior unhealthy alcohol users. Eighty-one percent (n = 87) of males were self-reported current drinkers, compared to 54% (n = 100) of the females (Table 1). Among those who reported prior unhealthy

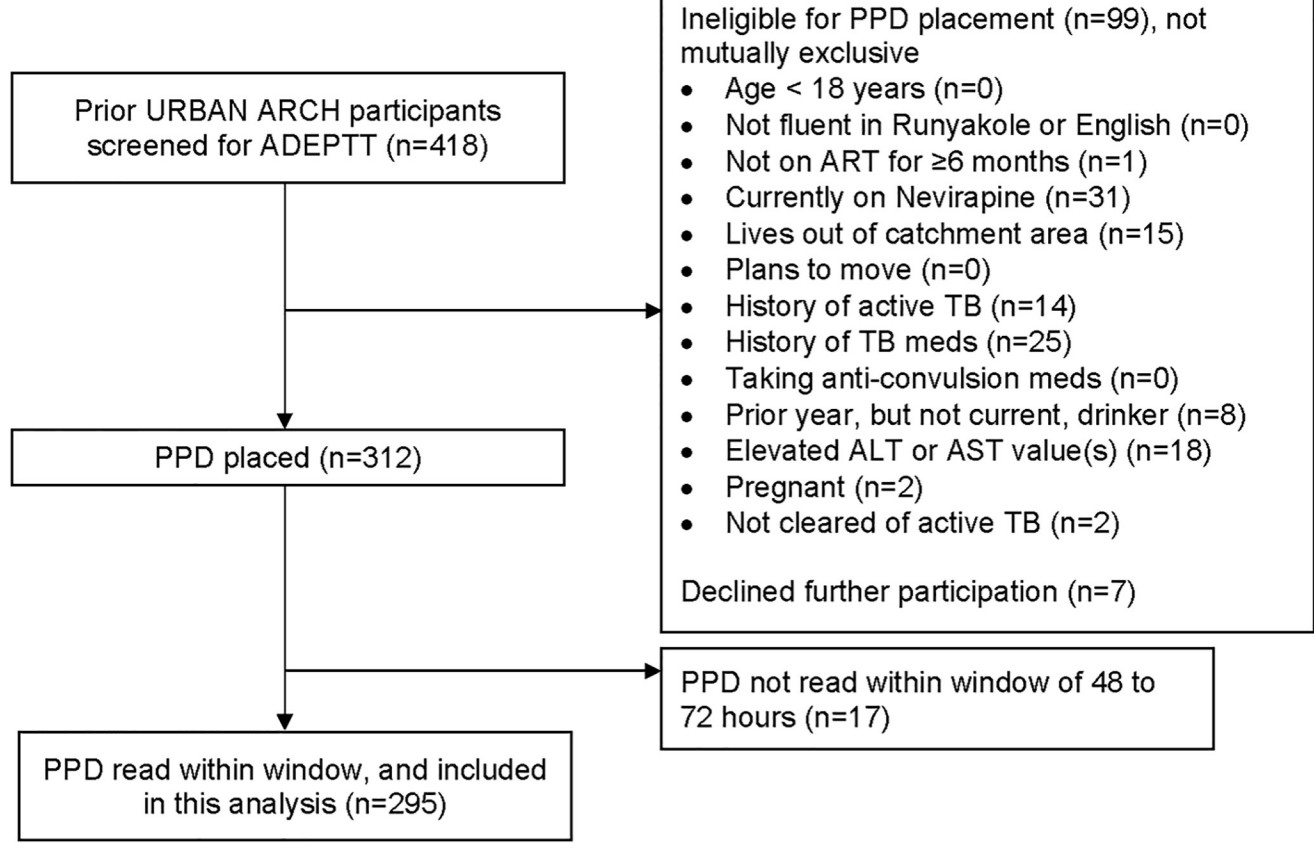

**Fig 1. Eligibility flow.**

alcohol use, 79% (n = 127) self-reported current drinking during ADEPTT screening, compared to 45% (n = 55) of those without prior unhealthy alcohol use.

## TST results

Of the 295 participants, 81 were TST positive (27.5%, 95% confidence interval [CI]: 22.6%, 32.9%). Among the current drinkers, 49 out of 187 (26%) were TST positive, while 32 out of 108 (30%) of abstainers were TST positive (Table 2). The unadjusted odds of a positive TST among current drinkers compared to abstainers was 0.84 (95% CI: 0.50, 1.43). The unadjusted odds of a positive TST among prior unhealthy alcohol users compared to participants with no prior unhealthy alcohol use was 1.08 (95% CI: 0.64, 1.82) (Table 2).

In multivariable analysis, the adjusted odds ratio [aOR] of a positive TST for current alcohol users compared to abstainers was 0.76 (95% CI: 0.41, 1.41) after controlling for gender, age, BMI, prior unhealthy alcohol use and history of smoking (Table 2). There was also no evidence of an association between positive TST and prior unhealthy alcohol use (aOR 1.14, 95% CI: 0.63, 2.07). The relationship between current alcohol use and a positive TST did not appear to differ by gender (p-value for interaction = 0.86): aOR: 0.68 (95% CI: 0.22, 2.05) for males and aOR: 0.89 (95% CI: 0.41, 1.91) for females (stratified analyses, Table 3).

## Discussion

We found that LTBI prevalence among HIV-infected persons on ART in south-western Uganda was 27.5% (95% CI: 22.6%, 32.9%). This was a little lower than the prevalence reported

**Table 1. Characteristics of eligible study participants in south-western Uganda, overall and by self-reported alcohol use at screening (n = 295).**

| | N (%) | Self-reported alcohol abstainers (at least 12 months) (n = 108) | Self–reported current (prior 3 months) alcohol users (n = 187) |
|---|---|---|---|
| TST result [a] | | | |
| TST-positive (≥5 mm induration) | 81 (27.5) | 32 (39.5) | 49 (60.5) |
| TST-negative (<5 mm induration) | 214 (72.5) | 76 (35.5) | 138 (64.5) |
| Sex [a] | | | |
| Male | 108 (36.6) | 21 (19.4) | 87 (80.6) |
| Female | 187 (63.4) | 87 (46.5) | 100 (53.5) |
| Age (median (IQR)) [b] | 38 (32–45) | 40 (32.5–48) | 37 (32–43) |
| <35 | 103 (34.9) | 33 (32.0) | 70 (68.0) |
| ≥35 | 192 (65.1) | 75 (39.1) | 117 (60.9) |
| Smoking history (lifetime) [c] | | | |
| Ever | 50 (17.3) | 15 (30.0) | 35 (70.0) |
| Never | 239 (82.7) | 88 (36.8) | 151 (63.2) |
| Body Mass Index (median (IQR)) [b] | 23.3 (20.8–27.5) | 23.5 (20.8–27.5) | 23.1 (20.8–27.6) |
| Low (<18.5) | 24 (8.5) | 9 (37.5) | 15 (62.5) |
| Normal (18.5–24.9) | 149 (52.8) | 54 (36.2) | 95 (63.8) |
| Overweight (25–29.9) | 70 (24.8) | 26 (37.1) | 44 (62.9) |
| Obese (> = 30) | 39 (13.8) | 14 (35.9) | 25 (64.1) |
| Marital status [b] | | | |
| Not married | 128 (43.8) | 53 (41.4) | 75 (58.6) |
| Married | 164 (56.2) | 54 (32.9) | 110 (67.1) |
| Prior unhealthy alcohol use [c] | | | |
| No | 123 (43.3) | 68 (55.3) | 55 (44.7) |
| Yes | 161 (56.7) | 34 (21.1) | 127 (78.9) |
| Duration of ART [b] | | | |
| < 1 year | 23 (7.8) | 8 (34.8) | 15 (65.2) |
| > = 1 year | 272 (92.2) | 100 (36.8) | 172 (63.2) |
| HIV viral load (median (IQR)) [b] | 75 (20–75) | 75 (20–75) | 75 (40–75) |
| < = 1000 | 287 (97.3) | 104 (36.2) | 183 (63.8) |
| >1000 | 8 (2.7) | 4 (50.0) | 4 (50.0) |

[a] ADEPTT Study screening data;

[b] ISS Clinic data;

[c] Prior URBAN ARCH study visit data.

in other LTBI based studies in SSA that ranged from 34–75% [4–6]. Differences may be attributed to variations in TB burden in different communities and differing TB infection control practices. At the ISS clinic from which research participants were recruited for this study, there is routine clinical screening for TB symptoms, isolation of TB suspects to a designated space, and expedited referral for TB investigation and treatment procedures for confirmed TB cases.

**Table 2. Unadjusted and adjusted Odds Ratios (OR) and 95% confidence intervals (95% CI) for TST-positive results among study participants who completed TST screening (n = 295).**

| | TST-positive [a] (n = 81) N (%) | TST-negative [a] (n = 214) N (%) | Unadjusted OR (95% CI) | p-value | Adjusted OR (95% CI) | p-value |
|---|---|---|---|---|---|---|
| Alcohol use–self-report at screening [a] | | | | 0.53 | | 0.39 |
| Abstainer (≥12 months) | 32 (29.6) | 76 (70.4) | 1.00 | | 1.00 | |
| Current drinker (≤3 months) | 49 (26.2) | 138 (73.8) | 0.84 (0.50, 1.43) | | 0.76 (0.41, 1.41) | |
| Sex [a] | | | | 0.24 | | 0.35 |
| Male | 34 (31.5) | 74 (68.5) | 1.00 | | 1.00 | |
| Female | 47 (25.1) | 140 (74.9) | 0.73 (0.43, 1.23) | | 0.73 (0.37, 1.42) | |
| Age [b] | | | | 0.73 | | 0.98 |
| <35 years | 27 (26.2) | 76 (73.8) | 1.00 | | 1.00 | |
| ≥35 years | 54 (28.1) | 138 (71.9) | 1.10 (0.64, 1.89) | | 1.01 (0.56, 1.80) | |
| BMI [b] | | | | 0.17 | | 0.19 |
| Low (<18.5) | 11 (45.8) | 13 (54.2) | 1.00 | | 1.00 | |
| Normal (18.5–24.9) | 35 (23.5) | 114 (76.5) | 0.36 (0.15, 0.88) | | 0.38 (0.15, 0.94) | |
| Overweight (25–29.9) | 19 (27.1) | 51 (72.9) | 0.44 (0.17, 1.15) | | 0.50 (0.18, 1.40) | |
| Obese (> = 30) | 11 (28.2) | 28 (71.8) | 0.46 (0.16, 1.34) | | 0.57 (0.18, 1.82) | |
| BMI (continuous) (median (IQR)) [b] | 23.0 (20.2–27.5) | 23.4 (21.0–27.6) | 1.00 (0.95, 1.05) | 0.96 | - | |
| Prior unhealthy alcohol use [c] | | | | 0.77 | | 0.66 |
| No | 34 (27.6) | 89 (72.4) | 1.00 | | 1.00 | |
| Yes | 47 (29.2) | 114 (70.8) | 1.08 (0.64, 1.82) | | 1.14 (0.63, 2.07) | |
| Smoking history [c] | | | | 0.30 | | 0.80 |
| Never | 64 (26.8) | 175 (73.2) | 1.00 | | 1.00 | |
| Ever | 17 (34.0) | 33 (66.0) | 1.41 (0.73, 2.70) | | 1.10 (0.53, 2.31) | |

[a] ADEPTT Study screening data;

[b] ISS Clinic data;

[c] Prior URBAN ARCH study visit data.

We found no differences between the odds of a positive TST among HIV-infected current alcohol users compared to abstainers in crude analyses or after controlling for age, gender, BMI, smoking history, and prior unhealthy alcohol use. This study provides descriptive information and the first estimates of LTBI among PLWH by current alcohol use. While there is literature on the association between heavy alcohol use, alcohol use disorders, and TB disease [9, 10, 14, 31, 32], this study evaluates the association between alcohol use and LTBI. It is possible that the higher rates of active TB disease referenced in unhealthy alcohol users may more likely be primary TB than reactivation given a higher likely exposure in bars and congested social drinking venues [11, 13]. However, it has also been observed that unhealthy alcohol consumption may disrupt some immune pathways and render reactivation to active TB disease more likely [11, 14]. Our data, which showed only the prevalence of latent TB infection, was not sufficient to allow us to examine these pathways. Further research in this area is needed.

We additionally found no relationship between prior unhealthy alcohol use and TST positivity. This result contrasts the higher odds of LTBI in drinkers in China and India [33, 34], but is consistent with results found in South Africa [6, 24]. This may suggest that 'any' current and/or prior unhealthy alcohol use may be playing a lesser role in determining the body's response to a tuberculin test among PLWH on ART. The findings may be influenced by the lack of information on the frequency, magnitude and timing of current alcohol use. Although

**Table 3. Adjusted Odds Ratios (OR) and 95% confidence intervals (95% CI) for TST-positive results among study participants who completed TST screening (n = 295), stratified by sex.**

| | MALES [a] (n = 108) | | FEMALES [a] (n = 187) | |
|---|---|---|---|---|
| | Adjusted OR (95% CI) | p-value | Adjusted OR (95% CI) | p-value |
| Alcohol use–self-report at screening [a] | | 0.49 | | 0.76 |
| Abstainer (≥ 12 months) | 1.00 | | 1.00 | |
| Current drinker (≤3 months) | 0.68 (0.22, 2.05) | | 0.89 (0.41, 1.91) | |
| Age [b] | | 0.10 | | 0.28 |
| <35 years | 1.00 | | 1.00 | |
| ≥35 years | 0.43 (0.16, 1.18) | | 1.50 (0.72, 3.13) | |
| BMI [b] | | 0.08 | | 0.43 |
| Low (<18.5) | 1.00 | | 1.00 | |
| Normal (18.5–24.9) | 0.37 (0.11, 1.21) | | 0.32 (0.07, 1.42) | |
| Overweight (25–29.9) | 1.20 (0.27, 5.39) | | 0.31 (0.07, 1.48) | |
| Obese (> = 30) | - | | 0.45 (0.09, 2.21) | |
| Prior unhealthy alcohol use [c] | | 0.58 | | 0.85 |
| No | 1.00 | | 1.00 | |
| Yes | 1.34 (0.48, 3.75) | | 1.07 (0.50, 2.29) | |
| Smoking history [c] | | 0.65 | | 0.93 |
| Never | 1.00 | | 1.00 | |
| Ever | 1.25 (0.47, 3.33) | | 0.95 (0.27, 3.32) | |

[a] ADEPTT Study screening data;

[b] ISS Clinic data;

[c] Prior URBAN ARCH study visit data

there may be a possibility that TST negative participants had not yet mounted a persistent immune response to mycobacterium tuberculosis bacteria, this may not be the case in our study participants who had been on antiretroviral therapy (ART) for more than 6 months and more than 95% of them were virally suppressed. Our work highlights the need to further examine how different levels of current alcohol use impact the risk of latent TB infection and progression to disease among drinkers in multiple settings.

This study has several limitations. The sample size was relatively modest due to multiple exclusion criteria for the parent study that screened out various groups of participants such as those with elevated alanine and aspartate aminotransferase—ALT and AST) more than 2 times the upper limit of normal. This was for the participants' safety as the INH to be given is potentially hepatotoxic. Since heavy alcohol drinkers have a higher risk of having elevated liver function tests, some might have been potentially eliminated at that point which may potentially affect generalizability. A comparative analysis between the participants included and those excluded from the analyses did not show any difference by alcohol consumption, age, gender, or BMI. In addition, self-reported alcohol consumption may be subject to recall bias. Our primary variable of current drinking was collected for study recruitment purposes, rather than to detect unhealthy drinking. Another limitation is that our additional variable to represent prior unhealthy drinking as well as some other variables of interest were collected 3–5 years prior. The median duration since last URBAN ARCH visit was 44.6 months (IQR: 36.2–53.9). Thus, the duration of time relative to TST assessment is inconsistent across the exposure variables and "lifetime" smoking and prior unhealthy alcohol use may be subject to misclassification due to the time lag. Although TST may not be a perfect test for LTBI, it is accepted by WHO [1] as one of the tests to detect latent TB. Both TST and interferon-gamma release assays

(IGRA) perform reasonably well in high-TB burden settings and correlate well with proxy measures of exposure to mycobacterium tuberculosis among the HIV-infected persons and contacts exposed to smear positive index cases [35]. Additionally, the concern in literature about TST's reduced sensitivity among the immunocompromised [27] was minimal in this study since the participants were on ART for more than six months and were clinically and virally stable. False negative results could also arise from the window period (2–8 weeks) before development of cell-mediated immunity estimated just after exposure to mycobacteria.

In summary, we found that LTBI by TST response among HIV-infected persons on ART in south-western Uganda was similar to global statistics though slightly lower than in prior SSA studies. Although prior unhealthy alcohol use existed in more than half the cohort, we did not detect an association between current alcohol use and LTBI. More research is needed to determine the effect of various magnitudes of alcohol drinking and LTBI as well as the mechanisms for increased TB disease among drinkers in the literature.

## Supporting information

**S1 File.**
(DOCX)

**S2 File.**
(DOCX)

**S3 File.**
(DOCX)

**S4 File.**
(DOCX)

**S5 File.**
(DOCX)

**S6 File.**
(DOCX)

**S1 Data.**
(XLSX)

## Acknowledgments

We acknowledge the ISS clinic staff at Mbarara Regional Hospital and the research assistants for ADEPTT clinical trial for the immense work collecting the data used.

## Author Contributions

**Conceptualization:** Winnie R. Muyindike, Robin Fatch, Debbie M. Cheng, Nneka I. Emenyonu, Julian Adong, Benjamin P. Linas, Karen R. Jacobson, Judith A. Hahn.

**Data curation:** Winnie R. Muyindike, Robin Fatch, Nneka I. Emenyonu, Christine Ngabirano, Julian Adong, Judith A. Hahn.

**Formal analysis:** Winnie R. Muyindike, Robin Fatch, Debbie M. Cheng, Judith A. Hahn.

**Funding acquisition:** Debbie M. Cheng, Judith A. Hahn.

**Investigation:** Winnie R. Muyindike, Robin Fatch, Christine Ngabirano, Julian Adong.

**Methodology:** Winnie R. Muyindike, Robin Fatch, Debbie M. Cheng, Nneka I. Emenyonu, Julian Adong, Judith A. Hahn.

**Project administration:** Winnie R. Muyindike, Nneka I. Emenyonu, Christine Ngabirano, Julian Adong, Judith A. Hahn.

**Resources:** Winnie R. Muyindike, Debbie M. Cheng, Nneka I. Emenyonu, Christine Ngabirano, Judith A. Hahn.

**Supervision:** Winnie R. Muyindike, Nneka I. Emenyonu, Christine Ngabirano, Julian Adong, Judith A. Hahn.

**Validation:** Winnie R. Muyindike, Robin Fatch, Debbie M. Cheng, Nneka I. Emenyonu, Christine Ngabirano, Julian Adong, Judith A. Hahn.

**Visualization:** Winnie R. Muyindike, Robin Fatch, Debbie M. Cheng, Nneka I. Emenyonu, Judith A. Hahn.

**Writing – original draft:** Winnie R. Muyindike, Robin Fatch, Judith A. Hahn.

**Writing – review & editing:** Winnie R. Muyindike, Robin Fatch, Debbie M. Cheng, Nneka I. Emenyonu, Christine Ngabirano, Julian Adong, Benjamin P. Linas, Karen R. Jacobson, Judith A. Hahn.

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
