## [Decision Letter · Decision Letter 0]

10 Mar 2020

PONE-D-19-35988

Tuberculin skin test positivity among HIV-infected alcohol drinkers on antiretrovirals in south-western Uganda

PLOS ONE

Dear Dr. Muyindike,

Thank you for submitting your manuscript to PLOS ONE. After careful consideration, we feel that it has merit but does not fully meet PLOS ONE’s publication criteria as it currently stands. Therefore, we invite you to submit a revised version of the manuscript that addresses the points raised during the review process.

We would appreciate receiving your revised manuscript by Apr 24 2020 11:59PM. To enhance the reproducibility of your results, we recommend that if applicable you deposit your laboratory protocols in protocols.io, where a protocol can be assigned its own identifier (DOI) such that it can be cited independently in the future. For instructions see: http://journals.plos.org/plosone/s/submission-guidelines#loc-laboratory-protocols

We look forward to receiving your revised manuscript.

Kind regards,

Nicky McCreesh

Academic Editor

PLOS ONE

Journal Requirements:

"I have read the journal's policy and the authors of this manuscript have the following competing interests: [Dr Debbie Cheng serves on Data Safety and Monitoring Boards for Janssen. The remaining authors have no conflicts of interest to declare."

We note that one or more of the authors are employed by a commercial company: Janssen

Reviewers' comments:

Reviewer's Responses to Questions

**Comments to the Author**

1. Is the manuscript technically sound, and do the data support the conclusions?

Reviewer #1: Yes

Reviewer #2: No

2. Has the statistical analysis been performed appropriately and rigorously? 

Reviewer #1: No

Reviewer #2: Yes

3. Have the authors made all data underlying the findings in their manuscript fully available?

Reviewer #1: No

Reviewer #2: No

4. Is the manuscript presented in an intelligible fashion and written in standard English?

Reviewer #1: Yes

Reviewer #2: Yes

5. Review Comments to the Author

Reviewer #1: 1. General comment: This manuscript generally adds to the body of knowledge on issue of LTBI. The authors need to explain the policy implications of their results given TB preventive therapy (TPT) is being recommended for all people with HIV. At a prevalence of 27.5% in the study population, do we seem to be recommending to 72.5% that do not need it. Or it is also possible that the 72.5% had not yet mounted a good immune response to reflect positivity?

2. Line 56: The authors should reference the latest global TB report (2019)

3. Line 64: Authors advised to cross check the latest incidence. It has come down lately

4. Line 166: Authors should indicate the IRB approval numbers

5. Line 173: The breakdown of the participants screened doesn’t total up to 106. Totals up to 118

6. Line 177: 418 minus total excluded= 300 not 312. Authors should crosscheck these figures in the section 172-181

7. Tables 2 and 3: The authors need to explain why they included all the factors in the multivariate analysis yet they were insignificant at bivariate? If these had biological plausibility, then they need to mention it and provide the appropriate references

8. Lines 226-237: Paragraph is too long. Could be split into 2 i.e. prevalence of LTBI and then the paragraph on odds of TST positivity given alcohol use history. The authors need to explain why there was no association between TST positivity among HIV infected current alcohol users and yet literature shows alcohol consumption is a big risk factor for TB. Is it possible that the TB among alcohol users is the primary TB instead of reactivation?

Reviewer #2: The introduction and abstract were generally clear and readable. A minor stylistic note is not to report results that are not findings of the study without reference, Since it is not practise to reference in abstracts perhaps it would be useful to say “there is evidence to suggest alcohol users have an increased risk of developing disease...” {leave it in intro line 72} There is a clear outline of the aim and synopsis of what is known and what is not known this is good. The purpose and hypothesis are also clearly defined. In the Methods section eligibility and sample are well described. The reliability and validity of using TST as a primary instrument are described in the discussion but should appear in the methods. It would be useful to have the complete definitions for “prior unhealthy” consumption. Systematic bias maybe introduced by only including TST screened for LTBI prevalence estimate. TST. There may be a potential systematic reason why these people did not present (too ill/heavy drinkers). The result tables are clear and readable. There is no stratification by alcohol use, the independent variable, it is an all or nothing measure. Therefore, the evidence is not sufficient for conclusions drawn in line 245 -247{This may suggest that alcohol use may be playing a lesser role in acquiring TB infection among PLWH on ART, compared to its role in TB disease progression}. These are merely measuring of association the lack of sample decreased the statistical value of these measures. For example, men report drinking (87%) but there are less men are in the study (37%). Drawing any links between alcohol use and TST response seem tenuous at best.

6. PLOS authors have the option to publish the peer review history of their article (what does this mean?). If published, this will include your full peer review and any attached files.

Reviewer #1: No

Reviewer #2: No

---

## [Author Response · Author response to Decision Letter 0]

27 Apr 2020

Re: Responses to Manuscript PONE-D-19-35988: “Tuberculin skin test positivity among HIV-infected alcohol drinkers on antiretrovirals in south-western Uganda”

Thank you all for the valuable time you spent reviewing our manuscript and the very thoughtful suggestions given to us. We also appreciate the opportunity to respond to the editor’s and reviewer’s recommendations. Below, please find a summary of the editor and reviewer’s comments and our responses to indicate how each point is addressed in the revised manuscript.

The Academic Editor gave the following communication:

Response: We have made adjustments to the following sections of the manuscript: The acknowledgments and the disclosure of conflict of interest. The details are included in our response to number 3 under Journal requirements.

To enhance the reproducibility of your results, we recommend that if applicable you deposit your laboratory protocols in protocols.io, where a protocol can be assigned its own identifier (DOI) such that it can be cited independently in the future.

Response: This is not applicable to our manuscript. This paper is based on a standard clinical screening procedure for the main study enrolling participants into a tuberculosis preventive therapy study using Isoniazid (INH). The clinical procedure conducted on each participant was a tuberculin skin test (TST) to detect latent tuberculosis infection (LTBI). For this clinical procedure, we used steps on a CDC manual chart for tuberculin testing (Reference #27 in the manuscript). 

The Journal requirements to be addressed included:

Response: We have read and abided by the style requirements for PLOS ONE including those for file naming. We have made some adjustments to the paper including: changes on the title page, changing manuscript text to double spacing and limiting the manuscript sections and sub-sections to 3 clear heading levels.

Response: This paper is based on screening steps before enrollment into the main tuberculosis prevention study. The questionnaires for the 5 screening steps in English and the one step which was translated into Runyankole (step 1) will be submitted as Supporting Information.

"I have read the journal's policy and the authors of this manuscript have the following competing interests: [Dr Debbie Cheng serves on Data Safety and Monitoring Boards for Janssen. The remaining authors have no conflicts of interest to declare."

We note that one or more of the authors are employed by a commercial company: Janssen

3.1 Please provide an amended Funding Statement declaring this commercial affiliation, as well as a statement regarding the Role of Funders in your study. If the funding organization did not play a role in the study design, data collection and analysis, decision to publish, or preparation of the manuscript and only provided financial support in the form of authors' salaries and/or research materials, please review your statements relating to the author contributions, and ensure you have specifically and accurately indicated the role(s) that these authors had in your study. You can update author roles in the Author Contributions section of the online submission form.

Response: We have updated the financial disclosure statement in the revised manuscript to read: All authors have no conflicts of interest to declare. (line 316)

Although Dr Debbie Cheng serves on Data Safety and Monitoring Boards for Janssen (as a consultant, not employee), after further evaluation, because Dr Cheng’s presence on a Data Safety Monitoring Board on a Janssen study for Crohn’s Disease is not in any way related to the TB preventive study from which this paper is written and does not in any way interfere with any steps in the research work from which this manuscript is derived, nor do we believe it interfered with the objective presentation of the peer review, we have hence removed that detail from this paper. 

The amended online Funding Statements may include: “The funders (NIAAA) provided support in the form of effort based salaries for the following authors [WM, RF, DC, NE, CN, JA, BL, KJ, JH (NIAAA) and NIAID to DC only, but did not have any role in the study design, data collection and analysis, decision on where to submit for publication, nor preparation of the manuscript. The specific roles of these authors are articulated in the ‘author contributions’ section.

3.2. Please also provide an updated Competing Interests Statement declaring this commercial affiliation along with any other relevant declarations relating to employment, consultancy, patents, products in development, or marketed products, etc. 

Response: We have updated the financial disclosure statement as mentioned in response to 3.1. We have also adjusted the acknowledgments section in the manuscript (lines 309-311) to read: This study was supported by Grants from National Institute on Alcoholism and Alcohol Abuse: U01 AA020776 and K24 AA022586 (PI: Hahn); U24AA020779 (PI: D Cheng) and NIH/NIAID P30AI042853 (support to D Cheng).

We apologize for the misquote in the previous paper. Debbie Cheng is not the PI of the NIH/NIAID P30AI042853 grant although she is supported by this grant whose PI is Dr S Cu-Uvin.

Response: We have consulted with the Institutional Review Boards and Committees that approved the main study. We find that there are no legal restrictions on sharing a de-identified data set once all the partners have agreed. We have prepared an anonymized dataset of the 295 participants on whom detailed analysis was done. 

Response: We have prepared an anonymized dataset of the 295 participants on whom detailed analysis was done. This will be submitted as a Supporting Information file. 

Response: The corresponding author has an ORCID iD in the Editorial Manager (0000-0002-6694-2645). We thank you for sharing the link to the elaborate video for instructions given. This has been very helpful.

Responses specific to Reviewer #1’s comments: 

1. General comment: This manuscript generally adds to the body of knowledge on issue of LTBI. The authors need to explain the policy implications of their results given TB preventive therapy (TPT) is being recommended for all people with HIV. At a prevalence of 27.5% in the study population, do we seem to be recommending to 72.5% that do not need it. Or it is also possible that the 72.5% had not yet mounted a good immune response to reflect positivity?

Response: As noted in the manuscript, the 27.5% prevalence is only somewhat lower than other estimates of LTBI prevalence in similar settings. Although there is a possibility that some of the 72.5% were actually latently infected with TB but had not yet mounted persistent immune response to mycobacterium tuberculosis bacteria, we think that this is likely to be minimal because our study participants had been on antiretroviral therapy (ART) for more that 6 months and more than 95% of them were virally suppressed. We have now added this to the discussion (lines 272-276). 

Because our findings are consistent with others’ and because our primary goal was to examine LTBI among those who consume alcohol compared to those who do not, there are no significant policy implications. We do not dispute the global call to expand access to TB preventive therapy because TB is the world’s top infectious disease killer from a single disease agent (WHO Global Tuberculosis Report 2019).

2. Line 56: The authors should reference the latest global TB report (2019)

Response: We have referenced the World Health Organization Global TB Report (2019) as Reference 1. We thank you for the advise to add this reference to our paper.

3. Line 64: Authors advised to cross check the latest incidence. It has come down lately

Response: The authors have checked the latest incidence from the WHO Global TB report 2019 and replaced the old incidence rates quoted in old paper with the latest incidence. This appears in lines 66 to 68 of the revised manuscript.

4. Line 166: Authors should indicate the IRB approval numbers

Response: Thank you for this communication. We have indicated the IRB approval numbers for the main study (lines 179-181 in the revised manuscript).

5. Line 173: The breakdown of the participants screened doesn’t total up to 106. Totals up to 118

Response: The authors appreciate this feedback. We have revised this section of the results to add a statement indicating that the reasons for excluding the 106 participants were not mutually exclusive (line 187). Some participants had more than one reason for exclusion and although counted under each exclusion criteria, the aggregated exclusion total counted each individual once. The full description is given in Figure 1.

6. Line 177: 418 minus total excluded= 300 not 312. Authors should crosscheck these figures in the section 172-181

Response: We apologize for the previous lack of clarity in line 177 of the old paper. With the response given in 5 above, we hope it is clearer how we came up with 312 as the participants who received TST (418 -106 = 312).

7. Tables 2 and 3: The authors need to explain why they included all the factors in the multivariate analysis yet they were insignificant at bivariate? If these had biological plausibility, then they need to mention it and provide the appropriate references

Response: We included several variables in the multivariate analysis model apriori (even when they were non-significant in the bivariate analysis) as we were concerned about confounding, which is not defined based on p-values. All of these variables have biological plausibility, that is, are documented high risk factors for progression to active TB (Refs 1,2,15), thus they were included regardless of statistical significance. 

8. Lines 226-237: Paragraph is too long. Could be split into 2 i.e. prevalence of LTBI and then the paragraph on odds of TST positivity given alcohol use history. The authors need to explain why there was no association between TST positivity among HIV infected current alcohol users and yet literature shows alcohol consumption is a big risk factor for TB. Is it possible that the TB among alcohol users is the primary TB instead of reactivation?

Response: We have split the previous long paragraph into 2 as advised. The first paragraph now occupies lines 244-251. The second paragraph has been expanded to include an explanation of the study findings compared to literature (lines 252-265).

It is possible that higher rates of active TB disease referenced in unhealthy alcohol users is more likely primary TB than reactivation given a higher likely exposure in bars and congested social drinking venues (Refs 11,13). However, it has also been observed that unhealthy alcohol consumption may disrupt some immune pathways and render reactivation to active TB disease more likely (Refs 11,14). Our data, which showed only the prevalence of latent TB infection, was not sufficient to allow us to examine these pathways. Further research in this area is needed.

Responses Specific to Reviewer #2’s comments: 

The introduction and abstract were generally clear and readable. A minor stylistic note is not to report results that are not findings of the study without reference, Since it is not practise to reference in abstracts perhaps it would be useful to say “there is evidence to suggest alcohol users have an increased risk of developing disease...” {leave it in intro line 72} 

Response: Thank you for this guidance. We have made the change advised in the abstract (line 31) but left it in the introduction, line 72 of revised manuscript).

There is a clear outline of the aim and synopsis of what is known and what is not known this is good. The purpose and hypothesis are also clearly defined. In the Methods section eligibility and sample are well described. The reliability and validity of using TST as a primary instrument are described in the discussion but should appear in the methods.

Response: We have moved the sentences describing the reliability and validity of using TST as a primary instrument from the discussion to the methods sections. These now appear in lines 131-136 within the methods section. 

It would be useful to have the complete definitions for “prior unhealthy” consumption. 

Response: We have expanded the definition for ”prior unhealthy” consumption in lines 147-151 of the methods section of the revised manuscript. 

The study defined prior unhealthy consumption (based on prior URBAN ARCH data) using the Alcohol Use Disorders Identification Test- Consumption (AUDIT-C), and modified those questions to ask about the period of the participant’s life during which they drank the most. Then, we used the normal cutoffs (>=3 for women, >=4 for men) to classify people as “prior unhealthy” drinkers. This description is in the “covariates” section.

Systematic bias maybe introduced by only including TST screened for LTBI prevalence estimate. There may be a potential systematic reason why these people did not present (too ill/heavy drinkers). 

Response: We note the valid concern raised. We may not rule out the systematic bias completely since the steps to determine eligibility were hinged towards the main TB preventive study. For instance, to be approached for screening, one had to be on ART for equal or more than 6 months. Given this pre-set eligibility, most participants were stable clinically, regardless of whether they were heavy drinkers or not. We screened participants who participated in prior URBAN ARCH studies of alcohol use which included all spectrum from mild to heavy alcohol users.

We also acknowledge the fact that the parent study eligibility criteria included having liver function tests (specifically alanine and aspartate aminotransferase - ALT and AST) less or equal to 2 times the upper limit of normal. This was for the participants’ safety as the INH to be given is potentially hepatotoxic. Since heavy alcohol drinkers have a higher risk of having elevated liver function tests, some might have been potentially eliminated at that point. We have added this to the study limitations lines 281-285.

The result tables are clear and readable. There is no stratification by alcohol use, the independent variable, it is an all or nothing measure. Therefore, the evidence is not sufficient for conclusions drawn in line 245 -247 {This may suggest that alcohol use may be playing a lesser role in acquiring TB infection among PLWH on ART, compared to its role in TB disease progression}. These are merely measuring of association the lack of sample decreased the statistical value of these measures. For example, men report drinking (87%) but there are less men are in the study (37%). Drawing any links between alcohol use and TST response seem tenuous at best.

Response: Thank you for this suggestion. In this study, we found no difference in LTBI prevalence in current or prior unhealthy alcohol users compared to abstainers. We have removed the bracketed statement above and replaced it with: this suggests that “any” current alcohol use and/or prior unhealthy alcohol use may be playing a lesser role in determining the body’s response to a tuberculin test among PLWH on ART (lines 268-270) . In another new study being done among HIV infected patients in care, our team of research partners will assess the prevalence and correlation of LTBI among current heavy alcohol drinkers. 

6. PLOS authors have the option to publish the peer review history of their article (what does this mean?). If published, this will include your full peer review and any attached files.

Response: This has been noted. Thank you very much for sharing this information.

Thank you again for the consideration and review of our manuscript. We hope that with these revisions, the manuscript is now suitable for publication in PLOS ONE.

Submitted by;

Dr Winnie R Muyindike 

Mbarara University of Science and Technology/Referral Hospital, Mbarara, Uganda 

Email: wmuyindike@gmail.com

Phone: +256 772 521619

---

## [Decision Letter · Decision Letter 1]

12 May 2020

PONE-D-19-35988R1

Tuberculin skin test positivity among HIV-infected alcohol drinkers on antiretrovirals in south-western Uganda

PLOS ONE

Dear Dr. Muyindike,

Thank you for submitting your manuscript to PLOS ONE. We provisionally accept the manuscript, but would like to give you the opportunity to address the final minor comments raised by the reviewer. Therefore, we invite you to submit a revised version of the manuscript that addresses the points raised during the review process.

We would appreciate receiving your revised manuscript by Jun 26 2020 11:59PM. To enhance the reproducibility of your results, we recommend that if applicable you deposit your laboratory protocols in protocols.io, where a protocol can be assigned its own identifier (DOI) such that it can be cited independently in the future. For instructions see: http://journals.plos.org/plosone/s/submission-guidelines#loc-laboratory-protocols

We look forward to receiving your revised manuscript.

Kind regards,

Nicky McCreesh

Academic Editor

PLOS ONE

Reviewers' comments:

Reviewer's Responses to Questions

**Comments to the Author**

1. If the authors have adequately addressed your comments raised in a previous round of review and you feel that this manuscript is now acceptable for publication, you may indicate that here to bypass the “Comments to the Author” section, enter your conflict of interest statement in the “Confidential to Editor” section, and submit your "Accept" recommendation.

Reviewer #1: All comments have been addressed

Reviewer #2: All comments have been addressed

2. Is the manuscript technically sound, and do the data support the conclusions?

Reviewer #1: Yes

Reviewer #2: Partly

3. Has the statistical analysis been performed appropriately and rigorously? 

Reviewer #1: Yes

Reviewer #2: I Don't Know

4. Have the authors made all data underlying the findings in their manuscript fully available?

Reviewer #1: No

Reviewer #2: Yes

5. Is the manuscript presented in an intelligible fashion and written in standard English?

Reviewer #1: Yes

Reviewer #2: Yes

6. Review Comments to the Author

Reviewer #1: (No Response)

Reviewer #2: Line 31: Comments have been addressed, but I would still add “….current evidence…”

Line 131-136: Comments have been addressed, but phrasing is still in discussion form, maybe best to some of it in the discussion as a limitation of this work.

(line 131) “TST may not be a perfect test for LTBI but it is widely accepted as a reasonable proxy [28] and”

AND

(line 134) “Both tests perform reasonably well in high-TB burden settings and correlate well with proxy measures of exposure [28].” I would move back to discussion and expand.

I would also move the sentences around in the method section (line 134/5) “TST may have reduced sensitivity in immune-compromised individuals such as… “ before the sentence starting (line 130) “A TST…”

281-285 comments have been addressed, but I would leave out (line 286) “However, the exclusions do not introduce a systematic bias that we see.”

line 245 -247 comments were addressed.

7. PLOS authors have the option to publish the peer review history of their article (what does this mean?). If published, this will include your full peer review and any attached files.

Reviewer #1: No

Reviewer #2: No

---

## [Author Response · Author response to Decision Letter 1]

7 Jun 2020

Responses specific to Reviewer #1’s comments: 

Comment to question, “Have the authors made all data underlying the findings in their manuscript fully available? reviewer 1 said No:

Response: We have prepared an anonymized dataset for all the 418 participants screened prior to conducting the tuberculin skin test (TST). The dataset that is attached on the 2nd resubmission includes those included for detailed analysis (295) and those excluded from the detailed analysis (123) who didn’t meet the criteria for inclusion for detailed analysis. This will be submitted as a Supporting Information file.

Responses specific to Reviewer #2’s comments: 

Line 31: Comments have been addressed, but I would still add “….current evidence…”

Response: We have added the word ‘current’ in the suggested position in line 31

Line 131-136: Comments have been addressed, but phrasing is still in discussion form, maybe best to some of it in the discussion as a limitation of this work. (line 131) “TST may not be a perfect test for LTBI but it is widely accepted as a reasonable proxy [28] and”

and (line 134) “Both tests perform reasonably well in high-TB burden settings and correlate well with proxy measures of exposure [28].” I would move back to discussion and expand.

Response: We have moved the suggested parts back in the limitations paragraph of the discussion and expanded. The previous line 131 and 134 now appear in the limitations under discussion, lines 292 to 299.

I would also move the sentences around in the method section (line 134/5) “TST may have reduced sensitivity in immune-compromised individuals such as… “ before the sentence starting (line 130) “A TST…”

Response: We have moved the suggested sentences around in the methods section. The previous lines 134/5 is now line 131. It has been inserted before the previous sentence which described the 5mm reading for a positive TST which is now line 132.

281-285 comments have been addressed, but I would leave out (line 286) “However, the exclusions do not introduce a systematic bias that we see.”

Response: We have deleted the line that previously read, “ However, the exclusions do not introduce a systematic bias that we see.” 

Thank you again for the consideration to process our manuscript for publication. We hope that with these revisions, the manuscript is now suitable for publication in PLOS ONE.

Submitted by;

Winnie R Muyindike

---

## [Editor Report · Decision Letter 2]

12 Jun 2020

Tuberculin skin test positivity among HIV-infected alcohol drinkers on antiretrovirals in south-western Uganda

PONE-D-19-35988R2

Dear Dr. Muyindike,

We’re pleased to inform you that your manuscript has been judged scientifically suitable for publication and will be formally accepted for publication once it meets all outstanding technical requirements.

Kind regards,

Nicky McCreesh

Academic Editor

PLOS ONE
---

## [Editor Report · Acceptance letter]

23 Jun 2020

PONE-D-19-35988R2 

Tuberculin skin test positivity among HIV-infected alcohol drinkers on antiretrovirals in south-western Uganda 

Dear Dr. Muyindike:

I'm pleased to inform you that your manuscript has been deemed suitable for publication in PLOS ONE. Congratulations! Your manuscript is now with our production department. 

Kind regards, 

on behalf of

Dr. Nicky McCreesh 

Academic Editor

PLOS ONE